# OpenReview forum: "SimpleDesign - A Joint Model for Protein Sequence and Structure Codesign"
_ICLR.cc/2026/Conference — Submitted to ICLR 2026_

### Official Review · Reviewer_uZGD · 2025-10-27

**Soundness:** 2
**Presentation:** 3
**Contribution:** 2
**Rating:** 4
**Confidence:** 4

**Summary:**

The paper introduces SimpleDesign, a protein sequence and structure co-design model, that leverages the Mixture of Transformers architecture. The model is trained end-to-end directly in the raw data space, explicitly avoiding the structural tokenization and multi-stage pipelines common in existing methods. The authors demonstrate strong, competitive results against leading existing co-design methods such as DPLM2 and ProtPardelle.

**Strengths:**

**Originality**: The paper does include a new architecture for co-design, based on the mixture-of-transformers. It is a simpler architecture, avoiding some of the very commonly seen modules in protein folding and structure designs (e.g. pairformer)

**Quality**: The performance is competitive with other methods, and comes out on top in some of the chosen metrics.

**Clarity**: The writing is clear and comprehensive. The introduction provides a good review that covers most of the relevant literature. The authors demonstrate a good understanding of the problem at hand, and some of the state-of-the art approaches. The figures, particularly figures 1 and 3 are very well designed, and clearly explained the model architecture, and the idea behind MoT.

**Significance**: The idea of simplifying architectures for protein design is important and useful.

**Weaknesses:**

While the paper has some strong points highlighted above, there are some major issues.

## Conceptual Points
- The paper focuses on unconditional protein design. While this is an interesting problem, there has been a lot of work towards protein design in the last few years (as the papers do highlight in their introduction), and a lot of this work has shown the difficulty of going from unconditional, to conditional protein design. At the same time, it is clear that there is little practical use of unconditional design models. For downstream scientific applications, designing generic proteins is not particularly helpful. Scientific use requires designing proteins to perform specific functions and/or bind to specific targets. Therefore, while these results would have merited conference acceptance 2/3 years ago, the competitive landscape means that unconditional design is (in this reviewer's opinion) not enough.
- SimpleDesign can jointly generate protein backbones and sequences. However (and again, the authors point this out in their introduction) there are now existing models that do all-atom generation. This is not only also co-design (as specifying all the atoms means implicitly defining the protein sequence) but also a much harder problem. It would have been interesting to see how SimpleDesign could combine the designed $C_{\alpha}$ backbone and sequence into an all-atom structure, and how that compares with existing all-atom models such as La Proteina.
- It is concerning to me that the model is trained only on the filtered AFESM dataset, a massive corpus of predicted/synthetic structures derived from AlphaFold and ESMFold synthetic structures (unless I have misunderstood). The authors do highlight throughout the paper, and clearly illustrate in figure 2, the relationship between co-design and folding. However, their training data comes entirely from folding models. This to me indicates that any issues or flaws of the folding models would also be learned by the model. It is not clear why experimental data for the PDB was not also considered during training, in addition to synthetic data.
- Finally, the paper talks extensively about the simplicity of their architecture. In my opinion, there are three possible arguments in favor of simplicity: It reduces computational cost, it might lead to improved performance (by adding the right inductive biases and/or preventing overfitting), or it is just for academic/learning purposes. It is unclear to me if the reason the authors wanted a simpler architecture was reduced cost, or if they actually expected improved performance because of the simpler architecture. In the latter case, it is not clear to me that the chosen architecture adds any inductive biases that would lead to improved performance (like symmetries in the 3D generation). It would be very useful to get some more clarity on why a practitioner would want to choose this simpler model, and not a bigger one, when they had to design proteins.

## Other issues
- In line 105 onwards, a number of models are mentioned to generate backbone atoms. These, however, include La Proteina, which is an all-atom method (it is in fact cited as such later in line 110). I would also suggest adding Latent-X to the all-atom methods cited.
- The results do not include any all-atom methods (which as discussed, are also doing co-design). In fact, given the choice of metrics, that results tables 1 and 2 could include 3D backbone design methods (RFDiffusion, FoldFlow, etc) combined with inverse folding (PMPNN for example). And, based on results from these backbone design papers, their numbers might be better than anything in tables 1 or 2.
- Performance is not bad, but there doesn't seem to be a unique design choice for SimpleDesign that actually surpasses all other models considered. This makes it harder to recommend the paper for publication

**Questions:**

- The model uses two times ($t$ and $t'$) to corrupt the sequence and structures. Would it be possible to use a unified time index, that is discretized for the sequences?
- In the results, the noise scale $\gamma$ seems to play a very important part, but there is no discussion in the main text about the choice of noise scale, or noise schedule. In fact, nowhere in the main text is it mentioned what $\gamma$ is. Given the importance of this parameter in the final results, it would be great to see more discussion on how it is chosen, and the tradeoff between TM score and diversity that comes from changing it.
- The co-design metric is, in fact, very similar to the designability metric often used in 3D backbone design, with the difference that backbone design requires inverse folding. This should be mentioned. Furthermore, the paper says that "one can use either scRMSD < 2 or scTM > 0.9", but it does not say which of these is actually used in the tables and figures.

---

> ### Author Response · Authors · 2025-11-20
> **Rebuttal response to Reviewer uZGD (1)**
>
> Thanks reviewer uZGD for their constructive feedback and careful assessment for our manuscript. Here are our response:
>
> ### Conceptual Points
> 1. [Concerns about unconditional protein design]. In the past few years, we have seen several works exploring unconditional protein design such as FrameDiff, Genie2, etc.. We agree that in practice (real-world protein-related research and development), conditions (such as functions, structural topology) are necessary to enable the steering and controlling the protein design. However, unconditional design can be used to evaluate the model’s base capacity for distribution modeling. In SimpleDesign, we choose unconditional co-generation as the benchmarking task, in order to reflect how SimpleDesign collectively models two different but highly relevant modalities.  In order to show the performance of SimpleDesign in conditional generation tasks (inverse folding and folding), we attach updated results by conducting inference of SimpleDesign on the evaluation of (1) folding for CASP14 targets (2) folding for CAMEO22 targets and (3) Inverse folding for CATH4.2 test targets, which are all common practice for benchmarking the respective tasks.
>
> | Task             | Dataset   | Metric           | Result (mean/median) |
> |------------------|-----------|------------------|---------------------------|
> | **Folding**      | CASP14    | TM-score         | 0.72 / 0.77               |
> |                  |           | RMSD (Å)         | 2.584 / 2.310             |
> |                  | CAMEO22   | TM-score         | 0.54 / 0.49               |
> |                  |           | RMSD (Å)         | 3.677 / 3.415             |
> | **Inverse Folding** | CATHv4.2 test     | Recovery (ARR)   | 0.359 / 0.369  |
>
> 2. [All-atom modeling]. Thanks for objectively raising this suggestion for all-atom modeling. We agree that all-atom modeling can be important because atom-level modeling enables us to predict or generate fine-grained interactions such as binder and ligand complexes. Here, we build SimpleDesign as an attempt in building a simplified framework for protein design with minimal requirement for inductive biases and would like to explore all-atom modeling as a valuable future direction. For example, aggregation of all-atom information instead of alpha carbon within the continuous token can be a feasible direction to further build upon SimpleDesign.
>
>
> 3. [Dataset usage]. Indeed, AFESM and SwissProt are all so-called “distillation” datasets, which means the structures are from a predictive folding model. These datasets are widely used in training modern folding models (eg., AF3, Boltz), and design models (eg., Proteina). There exists a possibility the folding model is not perfect enough to produce all reasonable data points. However, such distilled dataset is usually cleaner than raw experimental PDB dataset (e.g., no missing atoms/segments), which makes it widely adapted in training protein design models. Here, we follow the experience from previous works in building the data pipeline of SimpleDesign.
>
> 4. [Discussion on Simplicity]. Thanks for raising this valuable point. We want to clarify the motivation of designing our model with “simplicity” in mind (which is the key contribution in this work and has been supported by experimental results). (1) Simplicity does open up design choices approaches  to consider beyond tokenization or staged modeling of sequence and structures. (2) The performance of Simpledesign demonstrates that directly combining two completely different objectives for different modalities can also work well. (3) Simplicity allows practitioners in protein modeling and generation to readily borrow advanced techniques and architecture design from other domains (eg., text, image), and thus accelerating protein-related research. In summary, our contribution is not merely by introducing another new model which is simpler, but by advocating the possibility of simple modeling.

---

> > ### Author Response · Authors · 2025-11-20
> > **Rebuttal response to Reviewer uZGD (2)**
> >
> > ### Other issues:
> > 1. We acknowledged that Latent-X is a relevant protein design method and have properly cited it in our updated manuscript.
> >
> > 2. [Compared with backbone design+inverse folding]. We appreciate the reviewer’s insightful suggestion regarding staged co-design methods that factorize p(x,y)=p(x) p(y∣x), such as combining backbone-generation models (e.g., RFdiffusion, Proteina) with inverse folding models (e.g., ProteinMPNN). Indeed, these pipelines can provide strong baselines for co-generation, and recent work such as ESM3 (structure → sequence) highlights their potential. Our current study focuses on single-stage joint modeling of sequence and structure, and we did not perform a systematic comparison against the staged pipelines for SimpleDesign. We agree this is an important direction and will incorporate such comparisons in future work to provide a more comprehensive evaluation. We thank the reviewer for pointing out this valuable extension.
> >
> > 3. [Performance concerns]. We want to first highlight the goal of SimpleDesign is to demonstrate that a much simpler and non-tokenized formulation can achieve performance comparable to state-of-the-art tokenization-based approaches. SimpleDesign shows competitive performance one standard protein design benchmarks though it didn’t surpass all existing models on every metric. Achieving this level of competitiveness without relying on a discrete structure-tokenization pipeline is, we believe, a meaningful result: it suggests that discrete autoencoders may not be strictly necessary for high-quality sequence–structure co-design, and that a simpler, more scalable architecture can be a viable alternative foundation. We hope that this finding broadens the design space for future protein generative models rather than positioning SimpleDesign as a replacement for all existing methods.
> >
> > ### Questions:
> > Q1: [Explanation of time indices]. We do introduce two independent time indices “during training” to make the learning objective receive enough supervision from different combination of partially corrupted sequence and structure; however, during inference, we do adopt a simple path (where the indices t, t’ are tied to one time index) to perform co-design, as we have shown in Fig 2.
> >
> > Q2: [Explanation of Gamma]. We apologize for not making clear the noise scale in the main text. We will discuss more about the noise scale (and updated in the manuscript), as the scaling coefficient in the diffusion (random) term for flow inference. Thanks for pointing that out.
> >
> > Q3: [Explanation of co-designability]. We will more clearly mention the definition of co-designability and its relationship with widely used designability in the main text; for the designable definition, scRMSD and scTM, we actually report both in the main results (Tab 1-2) split by ‘/’. We will make this more clear to help readers to understand.

---

### Official Review · Reviewer_RfzU · 2025-10-30

**Soundness:** 3
**Presentation:** 2
**Contribution:** 3
**Rating:** 6
**Confidence:** 2

**Summary:**

Protein function depends on the interplay between amino acid sequences (discrete) and 3D structures (continuous), but existing generative models for protein co-design rely on multi-stage training (e.g., pretrained autoencoders for structure tokenization) that adds complexity and inefficiency. To address this, the authors propose SIMPLEDESIGN, a minimalist end-to-end multi-modal generative model that directly processes raw sequence and structure data without tokenization.

**Strengths:**

1. SIMPLEDESIGN breaks from prior multi-modal protein design paradigms by eliminating structural tokenization and multi-stage training—a core limitation of models like ESM3 and DPLM2.
2. The MoT architecture is also innovative: unlike fully decoupled or naive fused models, it explicitly models modality-specific signals (e.g., sequence symbolicity, structure geometry) while enabling cross-modal interaction, aligning with biological priors of sequence-structure coupling.

**Weaknesses:**

1 .SIMPLEDESIGN is evaluated exclusively on proteins of length 100–500, but many biologically critical proteins (e.g., fibrous proteins, multi-domain enzymes) exceed 500 residues. Additionally, it relies on 3D coordinates (Cα atoms) and secondary structure signals, making it unable to handle intrinsically disordered proteins (IDPs)—~30% of eukaryotic proteins that lack fixed structures but play key roles in signaling. This limits its applicability to the full proteome.
2. No experiments test if generated sequences fold into functional structures (e.g., enzyme catalytic activity, ligand binding).

**Questions:**

1. The MoT uses separate QKV projections for sequence and structure. Did the authors test if sharing some projections (e.g., value vectors) could reduce parameters without degrading performance?
2. Why it is name SimpleDesign？ why it is simple?
3. Ablate the number of modality-specific FFN layers (1 vs. 2 vs. 3) and report performance—this could reveal if 1 layer is sufficient to capture modality signals, reducing model size.
4. Collaborate with experimental teams to express 5–10 generated proteins (e.g., enzymes) in vitro and test for activity (e.g., hydrolysis assays)

---

> ### Author Response · Authors · 2025-11-20
> **Rebuttal response to Reviewer RfzU**
>
> We appreciate the reviewer for their constructive feedback and suggestions. We have categorized the questions / concerns into the following points:
>
> 1.  [Discussion on Simplicity] Here we elaborate our motivation and rationale behind the simplicity in SimpleDesign. (1) Simplicity in our model means that we build the architecture and training objective with minimal inductive bias but still are still capable of achieving competitive performance in protein design. SimpleDesign simply fuses the cross entropy for sequence tokens and flow matching loss for atom coordinates in a unified model and objective, removing the necessity of extra tokenization stage, allowing end-to-end learning and the use of general purpose transformers blocks in our network. (2) The benefits of developing a “Simple” approach is to allow researchers and practitioners to employ established architectures like transformers without further adaptation in the protein domain. Staying simple will make the exploration of alternative data or training recipes relatively more tractable and predictable, and easier to borrow modern advances from other domains such as text LLMs and images.
>
> 2. [Architecture ablations]. We appreciate the reviewer’s suggestion regarding partial parameter sharing and the depth of modality-specific FFN layers. Table 4 already compares the two extreme configurations—fully shared (standard Transformer) and fully modality-separated (MoT). Their performance is highly comparable, where MoT shows better designability and diversity trade-off, indicating that the model is not particularly sensitive to the exact degree of parameter sharing. This suggests that intermediate variants—such as sharing only the value projections or reducing modality-specific FFN depth—are likely to yield similar accuracy–diversity trade-offs while reducing parameter count. Due to computational constraints during the rebuttal period, we cannot run additional ablations, but we agree this is a valuable direction and will include such experiments in the camera-ready version to better characterize the spectrum between fully shared and fully separated architectures.
>
> 3. [Collaborating with wet-labs]. Thanks for the suggestion. We do agree that, especially for the protein design task, the evaluation in silico alone is not sufficient to reveal the utility of any proposed AI models. We conduct experiments and evaluate with computational surrogate metrics following existing practice, to probe the performance of SimpleDesign and compare with baselines as a starting point. We would like to extend our work by validating the performance on generating functional, real-world meaningful protein targets as important future collaboration work. We truly believe the effectiveness of the model should be finally justified by the real-world feedback. In response to this, we have incorporated your valuable suggestion into one discussion paragraph in the updated paper - Appendix C.

---

> > ### Comment · Reviewer_RfzU · 2025-11-26
> >
> > Thanks for the rebuttal.

---

### Official Review · Reviewer_5nV4 · 2025-10-31

**Soundness:** 2
**Presentation:** 3
**Contribution:** 2
**Rating:** 2
**Confidence:** 5

**Summary:**

SimpleDesign leverages a mixture of transformer architecture to directly learn protein co-design over the the raw structure and sequence space. A central focus of SimpleDesign is the avoidance of structural tokenization, latent representations, and attention pair bias commonly found in prior protein design works. SimpleDesign performs competitively to prior tokenization-based co-design models.

**Strengths:**

- Competitive performance to tokenization based methods. Compared to DPLM2 the model is competitive but does not require any tokenization. This represents another modality of taking off the shelf protein LLM's and extending them to generative tasks through a simple objective with a lot of data.
- Evidence shows that pure transformers be trained for de novo protein generation for both continuous and discrete data types.
-MoT architecture is a very fitting design choice to preserve scalability but along couple these conditional data types.

**Weaknesses:**

- Overall outside of the new architecture the technical novelty is low when compared to MultiFlow which uses a nearly identical training objective.
- I find the simplicity angle quite interesting but the tangible benefits are not reported (speed, size, data consumption/need, accuracy)
- Novelty seems really high accompanied with low diversity (Table 1). Is there any concern about memorization or overfitting?
- Missing benchmarks
  - MultiFlow is cited yet not compared to
  - Protpardelle (1-c) is an all atom model yet La-Proteina is cited and not compared to
  - Similarly P(all-atom), and PLAID are not referenced/compared to.
  - FoldFlow-2 would also be good comparison for backbone only and sequence conditioned generation
- Competitive performance to tokenization based methods but those methods are far below state-of-the-art as seen in P(all-atom) and La-Proteina which both can be evaluated in CA/seq co-design mode as Protpardelle and the recent Protpardelle 1-c were used.
- Initializing weights with an existing LLM (ESM2-650M) is not very different than using the embeddings themselves (PLAID) or training a separate autoencoder.

**Questions:**

- SimpleDesign seems to generate very non diverse structures and sequences with high similarity toward the training data yet does not have high co-designability nor designability. Any thoughts as to what could be the reason for this?
- What happens if the model is not initialized with ESM2 weights?
- How much does the dataset impact the results? If SimpleDesign was trained on just PDB does the simple architecture still generalize or is the simple architecture require large data?
- How is the speed and memory usage compared to prior methods? I would assume the simple architecture performs quire strongly here but it is not reported.
- MultiFlow being the first data space CA/seq co-design method also evaluated on forward folding and inverse folding. Does SimpleFold perform well in these cases?
- Backbone only numbers from Table 5 do not match the long length model reported for Proteina and Genie2.
- Any thoughts as to why the mmseqs sequence diversity is low for SimpleDesign in Figure 8?
- For the co-generated samples in Table 1 what is the amino acid distribution? Is there any over population of any of the amino acid types?

---

> ### Author Response · Authors · 2025-11-20
> **Rebuttal response to Reviewer 5nV4 (1)**
>
> We thank the reviewer 5nV4’s detailed feedback and suggestion in improving our manuscript. We want to answer and discuss on each point as follows:
>
> 1. [View of technical novelty]. We respectfully disagree with the reviewer on the evaluation of technical novelty as a ML research paper. The core message of this work is to explore the potential of a simple recipe for protein multimodal model training, where we use co-generation / design as an evaluation setup. In particular, we explore a capable methodology that shows strong departure from the existing work (e.g., MultiFlow and DPLM2). SimpleDesign shows competitive performance on standard protein design benchmarks while built upon simplified architecture and training objectives with minimal inductive biases.  We believe this can motivate more diverse research endeavors and make a valuable contribution to the community.
>
> 2. [Simplicity and efficiency]. We want to emphasize that simplicity does not imply computational “efficiency”. We highlight that the simplicity of our approach provides a more controllable approach to model design and iteration, and it allows co-design models to borrow methods for generative modeling in other domains like text-to-image generation. For example, since SimpleDesign is a pure flow matching generative model, we could borrow  distillation techniques that can largely decrease the number of timesteps in inference.
>
> 3. [Potential low-diversity or mode collapse]. We appreciate the reviewer’s concern regarding diversity. (1) We would like to clarify that diversity can behave differently depending on the metric used. Under the FS-Clus metric, our method indeed produces fewer distinct clusters than some baselines; however, under the TM-score diversity metric our method achieves competitive diversity. This indicates that our model does not actually collapse to a single dominant mode, as visualized in Figure 9. We want to further clarify that Foldseek measures structural similarity through the 3Di structural alphabet, which is more sensitive to fine-grained local structural variations, whereas TMscore focuses on global structural deviation after alignment. (2) Furthermore, we show in the appendix that the diversity–quality trade-off can be adjusted via inference parameters: increasing $\gamma$ shifts the model toward higher diversity at a controlled cost in reconstruction accuracy. While we do not claim to uniformly dominate all baselines, the results demonstrate that the method provides a controllable balance between quality and diversity instead of suffering from uncontrolled mode collapse.

---

> > ### Author Response · Authors · 2025-11-20
> > **Rebuttal response to Reviewer 5nV4 (2)**
> >
> > 4. Regarding the points of missing benchmarks, we have included the results of MultiFlow and La-proteina. We note that  due to the strong differences in training data, the baselines may not be directly comparable. Nevertheless, we include the reproduced results with official implementations as shown below. In particular, we didn't include baselines in our submission due to (1) MultiFlow use a ProteinMPNN-distillation data (where ProteinMPNN is also used in evaluation) which largely enhances the performance [1]. (2) La-proteina is concurrent work in parallel to SimpleDesign which is also under review at ICLR26. (3) For P(all-atom), we find that it cannot accommodate the commonly used GPU memory (up to 80GB)  during evaluation (for L=500) even with batch size = 1, so we still cited it as one of the relevant works.
> >
> > Table 1. Co-design Results (‘/’ split by scRMSD < 2A or scTM > 0.9)
> > | Method              | Co-design.    | TMscore div     | FS Clus. div    | Novelty         |
> > |---------------------|---------------|-----------------|-----------------|-----------------|
> > | Multiflow           | 0.762 / 0.804 | 0.3372 / 0.3362 | 0.5433 / 0.5249 | 0.8274 / 0.8257 |
> > | La-proteina (no-tri)| 0.7080 / 0.7420 | 0.3296 / 0.3262 | 0.596 / 0.5984 | 0.8109 / 0.8082 |
> > | La-proteina (tri)   | 0.77 / 0.792  | 0.3615 / 0.3615 | 0.3117 / 0.3114 | 0.8468 / 0.8471 |
> >
> > Table 2. Structure Design@1 Results  (‘/’ split by scRMSD < 2A or scTM > 0.9)
> > | Method              | PMPNN1        | TMscore div     | FS Clus. div    | Novelty         |
> > |---------------------|---------------|-----------------|-----------------|-----------------|
> > | Multiflow           | 0.856 / 0.894 | 0.3343 / 0.3347 | 0.5257 / 0.5324 | 0.8255 / 0.8240 |
> > | La-proteina (no-tri)| 0.84 / 0.864  | 0.3259 / 0.3263 | 0.607 / 0.6134  | 0.7983 / 0.7971 |
> > | La-proteina (tri)   | 0.842 / 0.876 | 0.3546 / 0.3542 | 0.3349 / 0.3562 | 0.8384 / 0.8369 |
> >
> > Table 3. Structure Design@8 Results  (‘/’ split by scRMSD < 2A or scTM > 0.9)
> > | Method              | PMPNN8        | TMscore div     | FS Clus. div    | Novelty         |
> > |---------------------|---------------|-----------------|-----------------|-----------------|
> > | Multiflow           | 0.952 / 0.982 | 0.3326 / 0.3322 | 0.5189 / 0.5234 | 0.8236 / 0.8224 |
> > | La-proteina (no-tri)| 0.95 / 0.966  | 0.3254 / 0.3241 | 0.6147 / 0.6066 | 0.7946 / 0.7955 |
> > | La-proteina (tri)   | 0.962 / 0.966 | 0.3516 / 0.3515 | 0.3763 / 0.3665 | 0.8321 / 0.8319 |
> >
> >
> > Table 4. Sequence Evaluation
> > | Method                | PPL     | PLDDT  | MMSeq div | Novelty |
> > |----------------------|---------|--------|-----------|----------|
> > | Multiflow            | 7.944   | 80.17  | 0.992     | /        |
> > | La-proteina (no-tri) | 11.399  |80.57| 0.988     | 0.4094   |
> > | La-proteina (tri)    | 11.893  | 83.49| 1.000         | 0.388    |
> >
> > 5. [Using ESM2's weight initialization] We follow previous practice to use the ESM2 initialization according to the findings in DPLM [1]. We would like to explore the effect of ESM2 pretraining in our multimodal settings, but the current budget cannot guarantee us a complete training from random initialization. We will update these in a later revision.
> >
> > 6. [Effect of training data]. This is an insightful question. In practice we find that the dataset quality can affect the evaluation metrics, and existing practices of protein design models do not follow a unified training data (MultiFlow uses PDB + distillation from ProteinMPNN); Proteina / La-proteina follows AFDB, etc. The motivation of choosing AFESM is because this includes the widest structure space among the publicly available dataset when we prepare the study. We cannot conclude based on the existing experiments whether the architecture indeed requires more data. Along with the initialization ablation, we will also explore the effect of dataset selection using our recipe, which is relatively less in the current protein design literature.
> >
> > 7. [Conditional tasks]. In order to show the performance of SimpleDesign in conditional generation tasks (inverse folding and folding), we attach updated results by conducting inference of SimpleDesign on the evaluation of (1) folding for CASP14 targets (2) folding for CAMEO22 targets and (3) Inverse folding for CATH4.2 test targets, which are all common practice for benchmarking the respective tasks.
> >
> > | Task             | Dataset   | Metric           | Result (mean/median) |
> > |------------------|-----------|------------------|---------------------------|
> > | **Folding**      | CASP14    | TM-score         | 0.72 / 0.77               |
> > |                  |           | RMSD (Å)         | 2.584 / 2.310             |
> > |                  | CAMEO22   | TM-score         | 0.54 / 0.49               |
> > |                  |           | RMSD (Å)         | 3.677 / 3.415             |
> > | **Inverse Folding** | CATH v4.2 test    | Recovery (ARR)   | 0.359/ 0.369  |

---

> > > ### Author Response · Authors · 2025-11-20
> > > **Rebuttal response to Reviewer 5nV4 (3)**
> > >
> > > 8. [Performance mismatch of Proteina and Genie2]. We thank the reviewer for pointing out this discrepancy. Our reported backbone-only results follow the default inference configurations provided in each official codebase and are aligned with the evaluation setup used for the other baselines and for SimpleDesign. The mismatch primarily arises because (i) we did not tune Genie2’s inference hyperparameters beyond the default settings, and (ii) for Proteina we used the publicly released 200M triangle model, rather than the long-chain model. To avoid ambiguity, we added all factual inference details to the Appendix. Due to limited computational resources, we were unable to rerun the full set of long-chain / hyperparameter tuning experiments during the rebuttal period, but we commit to updating the Genie2 and Proteina results in the next revision.
> > >
> > > 9. [Explanation of MMseqs diversity]. We thank the reviewer for the question regarding the low MMseqs sequence diversity in Fig. 8. The figure presents two checkpoints of SimpleDesign: (i) the pretrained model and (ii) the SwissProt-finetuned model, and the decrease in diversity is specific to the finetuned variant. We believe this behavior stems from the increased difficulty of jointly modeling sequence and structure during finetuning: unlike methods that operate on a single modality, SimpleDesign learns a coupled distribution, which can bias the model toward more conservative sequence choices when optimizing both modalities simultaneously during inference. A similar effect is observed in DPLM2[2], suggesting this may be a general phenomenon in multi-modal generative models rather than an issue unique to our architecture.
> > >
> > > 10. [Distribution over amino acid types] We have computed the statistical distribution of amino acid types for the generated samples and compared them with the PDB samples.
> > >
> > > | metric / AA       | A     | C    | D    | E    | F    | G    | H    | I    | K    | L     | M    | N    | P    | Q    | R    | S    | T    | V    | W    | Y    | Total  |
> > > |-------------------|-------|------|------|------|------|------|------|------|------|-------|------|------|------|------|------|------|------|------|------|------|--------|
> > > | SimpleDesign count      | 16420 | 1166 | 7564 | 11697| 4179 | 13249| 2595 | 9046 | 8087 | 15169 | 3413 | 4981 | 6893 | 5704 | 8855 | 7263 | 7309 | 11928| 861  | 3621 | 150000 |
> > > | SimpleDesign frequency  | 0.109 | 0.008| 0.050| 0.078| 0.028| 0.088| 0.017| 0.060| 0.054| 0.101 | 0.023| 0.033| 0.046| 0.038| 0.059| 0.048| 0.049| 0.080| 0.006| 0.024| 1.00  |
> > > | natural | 0.083  | 0.014 | 0.055 | 0.067 | 0.039 | 0.071 | 0.028 | 0.059 | 0.058 | 0.96  | 0.024 | 0.041 | 0.047 | 0.039 | 0.055 | 0.067 | 0.053 | 0.069 | 0.011 | 0.029 | 1.00  |
> > >
> > >
> > > Finally, we hope our answers and discussions address your questions!
> > >
> > > **References:**
> > >
> > > [1] Wang, Xinyou, et al. "Diffusion language models are versatile protein learners." arXiv preprint arXiv:2402.18567 (2024).
> > >
> > > [2] Wang, Xinyou, et al. "Dplm-2: A multimodal diffusion protein language model." arXiv preprint arXiv:2410.13782 (2024).
> > >
> > > [3] UniProtKB/Swiss-Prot protein knowledgebase release 2025_04 statistics, https://web.expasy.org/docs/relnotes/relstat.html

---

> ### Comment · Reviewer_5nV4 · 2025-11-22
> **Response to the Rebuttal**
>
> Thank you for the authors detailed response. Overall I agree the paper has some novel aspects but the poor empirical performance especially the novelty, and lack of support for claims outweighs this. The poor protein diversity and novelty are the limiting factors to raising the score as there is not any study of how or why this happens for SimpleDesign but not prior work.
>
> 1. Technical Novelty. I agree SimpleDesign is different from MultiFlow in that it uses simpler architecture but the architecture does not lend it self to much better practical use (accuracy/speed). Furthermore the use of copying the ESM2 weights and AFESM data with little practical gain compared to smaller architectures make the simplicity argument convoluted. The architecture may be simpler but the data use, training stages, and external model reliance should be factored in as well and ablated. A fair data comparison + LLM use ablation would highlight the architectural novelty that does exist much better and make the contributions more clear.
>
> If simplicity was the goal it would be important to see that on equal settings where simplicity can really shine. This is not clear in the current experiments. Also given the simple architectures training and evaluating on even larger proteins should be a place SimpleDesign could perform well.
>
> 2. Protein Novelty. SimpleDesign shows that nearly none of the proteins are novel when compared to the PDB. This seems evident of overfitting or lack of generalization.  This seems to be a flaw shared with DPLM2 that is also exhibited in both DPLM2 and SimpleDesign's sequence evaluation metrics.
>
> 3. Controllability. Simplicity does not provide controllability nor is any experiments of controllability shown. I understand the pitch but one should support this claim.
>
> 4. Lack of Diversity. Beyond Protpardelle most methods have similar trends with both TMscore and FS based diversity. The fact that when including fine grain details majority of proteins are seen as similar is a significant issue worth understanding. Temperature can control the tradeoff but the lack of diversity should not be overlooked as a metric flaw.
>
> 4/5. Benchmarks. Thank you for including the additional benchmarks. The lack of novelty of methods that initialize with EMS2 weights is worth further exploring. Further exploration of this correlation would strengthen the simplicity argument. PLAID should also be cirted and compared to.
>
> 8. The Genie2 and Proteina benchmarks are reproducible without hyperparameter search. The length of the training data should also be properly denoted as comparing in distribution to out of distribution skews the results.
>
> 9. Poor MMSeqs diversity and sequence novelty is worth exploring. While it is not due to the SimpleDesign architecture, why is exists between SimpleDesign and DPLM2 that use drastically different generative modeling frameworks is needed to better support the argument of the usefulness of simplicity.

---

> > ### Author Response · Authors · 2025-11-25
> >
> > Dear reviewer 5nV4,
> >
> > Thanks for your follow-up comments, and we appreciate your assessment.
> >
> > - [Protein diversity, novelty]. We acknowledge the low sequence novelty and MMSeqs diversity; this behavior appears across both SimpleDesign and DPLM2 and likely stems from the tighter constraints of joint sequence–structure modeling. We respectfully disagree that novelty (the avg similarity to known structures) is a well-defined metric that low novelty can also incur hallucination, and should not be over-weighted in evaluation. We agree SimpleDesign does not achieve SOTA in the aspect of FS-diversity, while the TM-score diversity remains competitive, indicating no structural mode collapse.
> > - [Technical novelty and simplicity]. Our main contribution is showing that a direct, non-tokenized architecture can match discrete‐autoencoder multi-modal methods (DPLM) without requiring structure tokenization. We agree that more ablations (e.g., pretrained vs. from scratch, data usage) will clarify the architectural role, and we will include these in the final revision.
> > - [Use of ESM2 weights or AFESM data].  We use ESM2 initialization primarily for efficiency and comparability, consistent with many recent protein models. We agree that an initialization ablation would strengthen the contribution, and we want to include a discussion and preliminary results in the fina revision. We have cited PLAID in the main text as well.
> > - [MMSeqs diversity shared with DPLM2]. The similar behavior between DPLM2 and SimpleDesign, despite different generative paradigms, suggests a modality-coupled effect rather than an architectural flaw. We will investigate additional investigations to clarify this shared phenomenon.

---

### Official Review · Reviewer_HGsG · 2025-11-02

**Soundness:** 2
**Presentation:** 3
**Contribution:** 3
**Rating:** 4
**Confidence:** 4

**Summary:**

This paper introduces SimpleDesign, a multi-modal generative model designed to jointly create novel protein amino acid sequences and their corresponding 3D structures. The core problem it addresses is that existing models often rely on complex, multi-stage training pipelines, particularly the use of autoencoders to tokenize continuous structural data into a discrete format. The authors hypothesize that this tokenization step is unnecessary. SimpleDesign instead operates directly on discrete sequences and continuous coordinates. It employs a Mixture-of-Transformer architecture that allows for modality-specific processing while still enabling joint self-attention across both sequence and structure data. The model is trained end-to-end with a simple, combined objective containing a discrete cross-entropy loss for the protein sequence and a continuous flow-matching regression loss for the protein structure. Experimental results show that SimpleDesign achieves competitive performance in co-generation and separate structure-only and sequence-only generation tasks, often outperforming its more complex, tokenized counterparts.

**Strengths:**

1. Proposing a novel SimpleDesign that eliminates the need for a separate structure tokenizer by operating directly on continuous 3D coordinates. This simplifies the entire training process into a single end-to-end objective.
2. SimpleDesign demonstrates a high co-designability score, surpassing other models. This suggests its joint objective and architecture are highly effective at capturing the correlation between the two modalities.
3. The authors leverage Mixture-of-Transformer (MoT) architecture, which effectively handles the multi-modal data by applying modality-specific and cross modal processing.

**Weaknesses:**

1. Across all unconditional generation tasks (co-design, structure-only, and sequence-only), the proposed method consistently exhibits low diversity. This suggests a significant risk of mode collapse.
2. The paper proposes using a Mixture of Transfomer architecture to better process modality-specific and cross modal information. However, the results in Table 4 are contradictory, showing that the MoT-based model achieves lower designability performance.
3. The evaluation is only limited to unconditional generation. The paper lacks results for conditional generation tasks (e.g., inverse folding and protein folding), which are more aligned with practical protein design scenarios and are essential for comprehensively demonstrating the proposed method's multimodal generation capabilities.
4. For the co-generation task, the authors should provide a comparison against more baselines such as [1][2][3][4].
5. SimpleDesign concatenates sequence and structure inputs along the sequence dimension, which is likely to introduce substantial computational overhead, especially when dealing with longer protein sequences.


[1] Generative Flows on Discrete State-Spaces: Enabling Multimodal Flows with Applications to Protein Co-Design

[2] P(all-atom) Is Unlocking New Path For Protein Design

[3] Elucidating the Design Space of Multimodal Protein Language Models

[4] La-Proteina: Atomistic Protein Generation via Partially Latent Flow Matching

**Questions:**

The paper models 3D coordinates directly rather than using structure tokenization. Could the authors provide a detailed discussion on the advantages of this direct modeling approach compared to using discrete structure tokens? Beyond the benefit of a simpler pipeline that already mentioned in the manuscript, are there other significant advantages to this design choice?

---

> ### Author Response · Authors · 2025-11-20
> **Rebuttal response to Reviewer HGsG (1)**
>
> We thank the reviewer HGsG’s constructive feedback and valuable thoughts! We answer to each concern and question as below:
>
> 1. [Potential low-diversity or mode collapse]. We appreciate the reviewer’s findings regarding diversity. (1) We would like to clarify that diversity scores can behave differently depending on the metric used. As shown in Table 1 & 2, under the FS-Clus metric, our method indeed produces fewer distinct clusters than some baselines; however, under the TM-score diversity metric, our method achieves competitive diversity. This indicates that our model does not actually collapse to a single dominant mode, as visualized in Figure 9. We want to further clarify that Foldseek measures structural similarity through the 3Di structural alphabet, which is more sensitive to fine-grained local structural variations, whereas TMscore focuses on global structural deviation after alignment. (2) Furthermore, we show in the appendix that the diversity–quality trade-off can be adjusted via adjusting inference parameters. Increasing $\gamma$ shifts the model toward higher diversity at a controlled cost in reconstruction accuracy. While we do not claim to uniformly dominate all baselines, the results demonstrate that the method provides a controllable balance between quality and diversity instead of suffering from uncontrolled mode collapse.
>
> 2. [Compare MoT <> Transformer architecture]. We thank the reviewer for pointing out the mixed results in Table 4. Indeed, the Mixture-of-Transformers (MoT) variant does not strictly improve the designability metric alone, but rather it allows to better separate modality-specific and cross-modal interactions and thereby enable a more favorable quality–diversity trade-off. As shown in Table 4 and Fig. 8, while the MoT variant attains slightly lower designability, it achieves higher structural diversity at comparable accuracy, and the diversity gain is consistent across multiple seeds.  This indicates that by introducing sparsity in Transformer architecture, MoT better models the distribution of both modalities rather than fitting into a single generation target that leads to low diversity.  Importantly, our goal is to demonstrate that MoT provides an alternative architecture with a different balance profile rather than claiming uniform superiority across all metrics.
> 3. [Conditional generation results] In order to show the performance of SimpleDesign in conditional generation tasks (i.e., inverse folding and folding), we provide updated results by conducting inference of SimpleDesign on the evaluation of (1) folding for CASP14 targets (2) folding for CAMEO22 targets and (3) Inverse folding for CATH v4.2 test targets, which are all common practice for benchmarking the respective tasks.
>
> | Task             | Dataset   | Metric           | Result (Ours / Baseline) |
> |------------------|-----------|------------------|---------------------------|
> | **Folding**      | CASP14    | TM-score         | 0.72 / 0.77               |
> |                  |           | RMSD (Å)         | 2.584 / 2.310             |
> |                  | CAMEO22   | TM-score         | 0.54 / 0.49               |
> |                  |           | RMSD (Å)         | 3.677 / 3.415             |
> | **Inverse Folding** | CATHv4.2 test    | Recovery (ARR)   | 0.359 / 0.369  |

---

> > ### Author Response · Authors · 2025-11-20
> > **Rebuttal response to Reviewer HGsG (2)**
> >
> > 4. [Comparison against more baselines] (MultiFlow, p(all-atom), Elucidating, La-proteina) Thanks for the suggestion of adding more baselines into the experimental table. Given the limited budget during rebuttal, we have accordingly added MultiFlow, La-proteina, including versions with triangle update (tri) and without triangle update (no-tri). It is noted that due to the essential differences in training data, the baselines may not be directly comparable. Nevertheless, we include the reproduced results with official implementations as shown below. In particular, we didn't include these baselines due to (1) MultiFlow use a ProteinMPNN-distillation data which largely enhances the performance [1], (2) La-proteina is concurrent work in parallel to SimpleDesign (also under review). We failed to include the p(all-atom) because the inference GPU memory exceeded our 80GB limit. For all mentioned methods, we will give them proper citation in the paper.
> >
> > Table 1. Co-design Results (‘/’ split by scRMSD < 2A or scTM > 0.9)
> > | Method              | Co-design.    | TMscore div     | FS Clus. div    | Novelty         |
> > |---------------------|---------------|-----------------|-----------------|-----------------|
> > | Multiflow           | 0.762 / 0.804 | 0.3372 / 0.3362 | 0.5433 / 0.5249 | 0.8274 / 0.8257 |
> > | La-proteina (no-tri)| 0.7080 / 0.7420 | 0.3296 / 0.3262 | 0.596 / 0.5984 | 0.8109 / 0.8082 |
> > | La-proteina (tri)   | 0.77 / 0.792  | 0.3615 / 0.3615 | 0.3117 / 0.3114 | 0.8468 / 0.8471 |
> >
> > Table 2. Structure Design@1 Results  (‘/’ split by scRMSD < 2A or scTM > 0.9)
> > | Method              | PMPNN1        | TMscore div     | FS Clus. div    | Novelty         |
> > |---------------------|---------------|-----------------|-----------------|-----------------|
> > | Multiflow           | 0.856 / 0.894 | 0.3343 / 0.3347 | 0.5257 / 0.5324 | 0.8255 / 0.8240 |
> > | La-proteina (no-tri)| 0.84 / 0.864  | 0.3259 / 0.3263 | 0.607 / 0.6134  | 0.7983 / 0.7971 |
> > | La-proteina (tri)   | 0.842 / 0.876 | 0.3546 / 0.3542 | 0.3349 / 0.3562 | 0.8384 / 0.8369 |
> >
> > Table 3. Structure Design@8 Results  (‘/’ split by scRMSD < 2A or scTM > 0.9)
> > | Method              | PMPNN8        | TMscore div     | FS Clus. div    | Novelty         |
> > |---------------------|---------------|-----------------|-----------------|-----------------|
> > | Multiflow           | 0.952 / 0.982 | 0.3326 / 0.3322 | 0.5189 / 0.5234 | 0.8236 / 0.8224 |
> > | La-proteina (no-tri)| 0.95 / 0.966  | 0.3254 / 0.3241 | 0.6147 / 0.6066 | 0.7946 / 0.7955 |
> > | La-proteina (tri)   | 0.962 / 0.966 | 0.3516 / 0.3515 | 0.3763 / 0.3665 | 0.8321 / 0.8319 |
> >
> >
> > Table 4. Sequence Evaluation
> > | Method                | PPL     | PLDDT  | MMSeq div | Novelty |
> > |----------------------|---------|--------|-----------|----------|
> > | Multiflow            | 7.944   | 80.17  | 0.992     | /        |
> > | La-proteina (no-tri) | 11.399  |80.57| 0.988     | 0.4094   |
> > | La-proteina (tri)    | 11.893  | 83.49| 1.000         | 0.388    |
> >
> > 5. [Computational overhead of sequence dimension]. The idea of  “concatenating” on the sequence length dimension follows the practice of DPLM models [2,3], which is set to be the tokenization-based counterpart of SimpleDesign. The computational overhead should be approximately similar as DPLM since it also uses a transformer
> >
> >
> > 6. [Advantages of direct modeling and “simple” design choices]. We appreciate the reviewer’s request for a deeper discussion of the rationale behind modeling 3D coordinates directly. Beyond the simplification of the modeling pipeline to make our model end-to-end trainable, this design choice provides two practical advantages. First, a direct continuous representation avoids potential information loss introduced by tokenization and discretization, which can be restrictive when modeling flexible or high-resolution structural variations. Also, training additional tokenizers introduces extra complexity in optimization and computational overhead, which is avoided in end-to-end frameworks like SimpleDesign. Second, using a coordinate-level parameterization allows the architecture to remain fully compatible with modern implementations of general purpose transformer blocks that are efficient and scalable, like FlashAttention. This formulation offers a more extensible foundation for future large-scale or multimodal protein generative models.
> >
> >
> > [1] Campbell, Andrew, et al. "Generative flows on discrete state-spaces: Enabling multimodal flows with applications to protein co-design." arXiv preprint arXiv:2402.04997 (2024).
> >
> > [2] Wang, Xinyou, et al. "Diffusion language models are versatile protein learners." arXiv preprint arXiv:2402.18567 (2024).
> >
> > [3] Wang, Xinyou, et al. "Dplm-2: A multimodal diffusion protein language model." arXiv preprint arXiv:2410.13782 (2024).

---

### Meta-Review · Area_Chair_LQHn · 2026-01-01

**Summary:**

SimpleDesign proposes an end-to-end multimodal protein co-design model that jointly generates sequence (cross-entropy) and backbone structure (flow matching) directly in raw space using a Mixture-of-Transformers architecture, avoiding structural tokenization. While results are competitive with some tokenization-based baselines, reviewers raised substantial concerns about limited novelty vs. closely related work, unclear practical benefit of the “simple” formulation, and problematic diversity/novelty behavior; overall, the evidence is not yet strong enough for acceptance.

Pros
* Clean end-to-end formulation (no structure tokenizer) and straightforward objective.
* Competitive performance on selected co-design metrics vs. some tokenization-based baselines.
* Rebuttal adds missing baselines and (some) conditional task results.

Cons
* Novelty and positioning remain unclear relative to MultiFlow / related pipelines; limited ablations (ESM2 init, data choice, architecture choices) weaken claims.
* Persistent concerns about low diversity / low novelty and potential overfitting/memorization; trade-off explanations are not fully convincing.
* Practical value of “simplicity” (speed, memory, scalability, controllability) is asserted but not demonstrated with direct measurements.

**Reviewer Concerns:**

Addressed (partially): (i) Added additional baselines (MultiFlow, La-Proteina variants) and clarified some benchmarking details; However these additional experiments do not show a clear advantage of the proposed method  (ii) responded to conditional-generation criticism by adding folding/inverse-folding numbers; (iii) clarified diversity metric mismatch (FS-Clus vs TM-score) and that inference parameters can adjust the quality–diversity trade-off.

Still outstanding: (i) Core novelty/simplicity claims still lack supporting ablations (ESM2 init vs scratch, AFESM vs PDB/other data, parameter-sharing spectrum); (ii) low novelty / low (fine-grained) diversity remain a central unresolved limitation without a clear diagnostic; (iii) no concrete evidence for efficiency/controllability claims (runtime/memory/timesteps, scaling to longer proteins); (iv) broader and fairer comparisons to strong staged pipelines and stronger SOTA setups remain missing.

**Reviewer Scores:**

Generally, given the initial concerns and ratings, a substantial shift in the overall scores appears unlikely:
* 5nV4 explicitly stated they would not increase their score, and their core concerns, particularly around limited novelty and low diversity in the generated proteins, remain unresolved.
* RfzU, while the most positive reviewer, does not present a sufficiently strong or compelling case to support acceptance.
* HGsG could potentially raise their score from 4 to 5 in light of the additional experiments, although lingering concerns about diversity and inconsistencies in the MoT ablation results may still limit enthusiasm.
* uZGD might also consider increasing their score from 4 to 5 given the added experiments, but concerns regarding the emphasis on unconditional generation and reliance on synthetic training data persist.

---

### Decision · Program_Chairs · 2026-01-26

Reject